# Polymer Composite Materials Based on Polylactide with a Shape Memory Effect for “Self-Fitting” Bone Implants

**DOI:** 10.3390/polym13142367

**Published:** 2021-07-19

**Authors:** P. A. Zhukova, F. S. Senatov, M. Yu. Zadorozhnyy, N. S. Chmelyuk, V. A. Zaharova

**Affiliations:** 1National University of Science and Technology “MISIS”, Leninskiy pr. 4, 119049 Moscow, Russia; Senatov@misis.ru (F.S.S.); priboy38@mail.ru (M.Y.Z.); nellichmelyuk@yandex.ru (N.S.C.); 2A.N. Kosygin Russian State University, St. Sadovnycheskaya 33/1, 115035 Moscow, Russia; vasilinaqss@gmail.com

**Keywords:** polylactide, polycaprolactone, shape memory, self-fitting, implants, recovery stress

## Abstract

The development of adaptive medical structures is one of the promising areas of bioengineering. Polymer composite materials based on polylactide (PLA) are interesting not only for their properties, such as biocompatibility, mechanical properties, biodegradation, and convenience of use, but also for demonstrating shape memory effect (SME). In this study, reducing the activation initiation temperature and the SME activation energy was achieved by forming a composite based on PLA containing 10% poly (ε-caprolactone) (PCL). The effect of the plasticizer on the structure, mechanical properties, and especially SME of the composite, was studied by DSC, SEM, FTIR spectroscopy, compression tests, and DMA. By varying the composition, the beginning of the SME activation was reached at 45 °C, and the apparent activation energy of the process decreased by 85 kJ/mol, ensuring safe and effective use of the material as a precursor for temporary self-fitting scaffolds for reconstructive surgery.

## 1. Introduction

In modern reconstructive surgery, polymer composite materials are increasingly used due to the combination of high biocompatibility and various functional properties; yet some properties of biomedical polymers have been insufficiently studied. The shape memory effect (SME) in such materials is particularly promising and extremely interesting for different uses. However, calculating, programming, and exploiting this effect are complex tasks that require a deep study of the molecular structure and characteristics of individual materials and their combinations [1]. 

Within the scope of medical applications, SME is of high interest in reconstructive surgery [2]. Recent research shows a significant step towards using polymer-based implants to replace bone tissue. Based on SME, it is possible to perform “self-fitting” of the implant to solve the problems of implant placement and its congruence [3]. For these purposes, a polymer scaffold was compressed at a temperature above [4], below [5], or near the SME activation temperature and cooled without removing the load to fix the temporary shape. The activation of “self-fitting” occurs directly in the recipient’s body by one of the known methods, e.g., a direct method—heating or using additional tools—magnetic particles and fields, ultrasonic treatment [6,7,8,9,10].

SME in polymer materials occurs due to conformational entropy and internal energy recovery after deformation [11,12]. SME is activated at temperatures that characterize changes in the condition of the material—glass transition and melting temperatures. The deformation of an amorphous polymer at a certain temperature, which characterizes the material, leads to the polymer chains stretch into more elongated conformations. This reduces the number of available chain conformations, which is energetically unprofitable [1]. When the load is removed, the shape is recovered based on the spatial location of the chain sections. Several parameters can describe this effect: activation temperature, activation energy, recovery stress and strain, recovery ratio, and rate. These estimated parameters can be analyzed by DSC and DMA methods [13].

Polylactide (PLA) is a biocompatible, biodegradable thermoplastic polymer, one of the most widely used polymers in such areas as biodegradable packaging and disposable products, electronic devices, construction, and especially biomedicine [14,15]. In addition, it has a strongly marked shape memory effect [2,16,17], which allows it to be used as a base matrix of a composite material for “self-fitting” implants [18]. In this material, SME is activated at the glass transition temperature (T_g_ = 55–65 °C). At this temperature, the polymer passes into a viscoelastic state, which means that the polymer chains acquire greater mobility, and the energy exceeds the activation energy required for SME implementation [19]. 

This paper aims to reduce the activation temperature of these processes for the safest use of the material as a precursor for the implant. In this case, the glass transition temperature and the threshold activation energy depend on the polymer’s molecular structure, precisely, on the molecular weight, the degree of crystallinity, and the level of chain entanglement. 

SME can also be controlled by introducing various plasticizers and dispersed fillers [20,21]. A decrease in the initial activation temperature of this effect may trigger an additional phase switch, such as a polymer with phase transitions occurring in other temperature ranges. The most used polymers are polyethylene glycol (PEG) [22], thermoplastic polyurethane (TPU) [23,24], polyhydroxyalkanoates (PHA) [25,26,27], poly(ε-caprolactone) (PCL) [28], and their combinations.

In this study, the PCL melting process as a trigger for lowering the SME activation temperature in the PLA matrix was used [9]. The PCL melting temperature is in a range of Tm = 60–65 °C, and the glass transition temperature is below zero (T_g_ = −60 °C). These two polymers are immiscible, and they do not interact with each other and rarely form any bonds between chains without copolymerization [29,30,31]. However, PCL inclusions in the PLA matrix affect the transition temperature to the viscoelastic state of the blend.

The studied biocompatibility of PLA and PCL polymers makes them useful candidates to be used in implantology. The shape memory effect in this area is relevant for optimizing surgery in view of the phenomenon of implant self-installation [32]. The calculation of the composition and the analysis of the composite’s structure should provide the necessary mechanical and thermal properties for use in the body.

## 2. Materials and Methods

### 2.1. Preparation of Test Specimens

The basis of the polymer composite material was a polylactide (PLA) (molecular weight of 110 kg/mol, Ingeo 4032D, Natureworks LLC). To optimize the use of SME of the composite, poly(ε-caprolactone) (PCL) was added in an amount of 10 wt.%. The molecular weight of the PCL is 70 kg/mol. This value was obtained by viscometry using the Mark–Hauwink constant k = 1.298 × 10^−4^; a = 0.828 (chloroform, 30 °C) [16].

The extrusion method was used to produce a composite polymer material using a two-screw extruder HAAKE MiniLab II micro compounder. The heating temperature of the operating section was 180 °C. The PLA and PCL pellets were gradually placed in the hot section in a ratio of 9:1. After loading, the polymer blend was homogenized for 20 min at 37 rpm. The filament then went out through the extrusion nozzle under additional pressure. The thickness of the filament was 1.5–1.7 mm.

The samples for shape memory evaluation by DMA were manufactured in the shape of plates 35 mm × 5 mm × 1 mm. The samples were obtained by hot molding (preload 2800 kPa, T = 200 °C, 45 min) of pre-milled polymer blend filaments. For the recovery stress/strain tests, the samples were deformed by 100% using a thermostat with a temperature of 80 °C. For the mechanical testing, this method was used to obtain cylindrical samples with a diameter of 13 mm and a height of 24–26 mm (preload 115,000 kPa, T = 200 °C, 75 min).

The cylindrical samples for the compression tests with a cross-section diameter of 13 mm and a height of 24–26 mm (ISO 604:2002) were obtained by a hot molding method.

To demonstrate the shape memory effect, a 40 mm × 40 mm plate was prepared. It fixed a temporary shape in the form of a twisted tube. The plate was heated in a metal container in the air at a temperature of 50 °C. 

### 2.2. Characterization 

#### 2.2.1. Structure

FTIR spectroscopy (Nicolet 380 spectrometer within a wave range 650–4000 cm^−1^) was used to evaluate chemical bonds in the polymer blend by identifying various functional groups.

The bulk samples were frozen in liquid nitrogen (T = 77 K) for 1 h and fractured to study a quasi-brittle fracture surface using a scanning electron microscope (Tescan Vega 3). Carbon deposition (layer 10–20 nm, SPI-MODULE carbon coater (SPI Inc., Lakewood, WA, USA)) was used to remove charge from the non-conducting samples.

The DSC method (NETZSCH DSC 204 F1 calorimeter, Germany) was used to determine the phase transitions of the composite material. Test mode: heating from 30 °C to 210 °C, holding at 210 °C for 5 min, cooling to 30 °C, reheating to 210 °C at a rate of 10 °C/min. 

#### 2.2.2. Mechanical Properties

The mechanical tests in compression were performed using a universal testing machine Zwick/Roell Z 020 (Zwick/Roell Group, Ulm, Germany). The measurements were carried out with a preload of 0.5 MPa at room temperature with a loading speed of 10 mm/min (ISO 604:2002). 

#### 2.2.3. Shape Memory Properties

The dynamic mechanical analysis (DMA) was performed using DMA Q800 mechanical analyzer (TA Instruments, New Castle, DE, USA). The temperature dependence of the elastic modulus, loss modulus, and *tan δ* was analyzed using the dual cantilever. The measurement was conducted in a temperature range of 26–70 °C with a heating rate of 2 °C/min. The oscillation strain amplitude was 0.1% at a frequency of 1 Hz. Deformed by 100% in length, the samples were tested to estimate such parameters as recovery stress and recovery strain. In the first case, the samples were fixed on both sides to determine the recovery force. In the second case, one of the clips was movable to evaluate the change in the length of the sample. 

To calculate the apparent activation energy of the SME, curves of *tan δ* dependence on temperature were obtained for different frequency values (1, 3, 5, 7, and 10 Hz). Based on the fact that the maximum of the curve corresponds to the activation temperature, graphical calculations were performed using the Arrhenius plot:(1)$$\ln f=\ln\;A-\frac{∆E_{a}}{RT},$$
where *f* is the frequency, Hz; A is the pre-exponential factor; Δ*E_a_* is the apparent activation energy of the SME, J/mol; *R* is the universal gas constant (8.31 J·mol^−1^·K^−1^).

The trend line constructed from the obtained points sets the slope, the tangent of which makes it easy to get the value of the apparent activation energy:(2)$$∆E_{a}=-R\left(\frac{d(\ln f)}{d\left(\frac{1}{T}\right)}\right)$$

#### 2.2.4. Cytotoxicity Assay

The SC1 cell line (ATCC) was cultured in DMEM (Dulbecco’s modified eagle medium) supplemented with 10% fetal bovine serum (FBS) and 2-mM L-glutamine (Gibco, Carlsbad, CA, USA). The cells were maintained at 37 °C in a humidified incubator MCO-18AC (Sanyo, Osaka, Japan) supplied with 5% CO_2_. After attaining 80% confluence, the cells were harvested with TrypLE (Gibco) and sub-cultured 1:8. The cell cultures were tested for the absence of mycoplasma.

Before the cytotoxicity test, the material samples were washed twice with 70% ethanol, then 3 times with DPBS (Dulbecco′s Phosphate Buffered Saline, Gibco, Waltham, MA, USA) and exposed to UV for 45 minutes.

The cytotoxicity of the samples was tested using the CellTiter 96 MTS 20 (3-(4,5-dimethylthiazol-2-yl)-5-(3-carboxymethoxyphenyl)-2-(4-sulfophenyl)-2H-tetrazolium, inner salt) cell proliferation assay kit (CellTiter 96 AQueous One Solution, PROMEGA, Madison, WI, USA). Samples were placed in wells of 96-well plate (Eppendorf, Hamburg, Germany). Then a suspension of cells at 7 × 103 cells per well were plated in 200 µL of growth media in 96-well plates and incubated in a 5% CO_2_ incubator for the first 48 h without treating. Cells incubated without samples and with the culture medium were used as negative controls. After 48 h of incubation, the media were changed (100 µL), and 20 µL of the MTS reagent was added into each well. After 4 h of incubation with the MTS reagent, the liquid was collected and transferred to empty wells and the absorbance was measured on a Thermo Scientific Multiskan GO microplate spectrophotometer (Waltham, MA, USA) at 490 nm. The results were used to construct a graph with GraphPad software. All tests were performed in triplicates. All data are displayed as mean ± SD of three replicates.

## 3. Results and Discussion

### 3.1. Analysis of Material Composition and Structure

The FTIR spectrum of the composite blend of PLA and PCL is shown in Figure 1. The region of 2850–3000 cm^−^^1^ includes three bands of 2998, 2946, and 2854 cm^−^^1^, which relate to symmetric/asymmetric bending of groups −C−H in PLA and to −CH_3_ stretching. At the same time, these spectrum peaks characterize asymmetric and symmetric −CH_2_ stretching in PCL at 2946 and 2865 cm^−^^1^, respectively. In the other part of the spectrum, the peaks can be attributed to the following vibrations: −C=O stretching (1751 cm^−^^1^), −C−H_3_ bending (1360 and 1454 cm^−^^1^), −CH3 bending, and −C−COO stretching (1081 and 1266 cm^−^^1^) for PLA. The following bands belong to PCL: C=O stretching (1724, 1182 cm^−^^1^), valence vibrations C−O−C (869, 1043, and 1109 cm^−^^1^), C=C stretching (1286 cm^−^^1^) [28,33].

The microphotographs of the quasi-brittle fracture surface of a PLA/PCL polymer blend were obtained using scanning electron microscopy (Figure 2). Cavities and spherical inclusions of less than 5 microns were found on the fracture surface. This surface relief is most likely due to the fact that PCL coalesces into small droplets and appears as an inclusion in the PLA matrix [28,30,34,35]. Due to the mechanical action, cavities where these droplets were located can be observed at the fracture surface, and in some places, inclusions are visible. Their formation proves the immiscibility of the two polymers.

The DSC method was used to evaluate thermal transitions in the polymer blend. The method was performed in three stages: first heating, cooling, and second heating. The study results are shown in Figure 3. 

The first heating eliminates the sample’s thermal history. The melting peaks of PCL and PLA are found at 57.5 and 152.3 °C, respectively [9,28]. At 98.4 °C, an exothermic effect of cold crystallization of PLA is observed, which indicates that PLA does not completely crystallize when the samples are obtained at temperature treatment. 

During the first heating, the melting peak of PCL and the glass transition of PLA overlap in a temperature range of about 45–62 °C, so it is impossible to determine the values accurately. When reheated, two peaks of 55.5 and 60.5 °C are visible due to the changes in the crystal structure of polymers due to cooling at a sufficiently high speed [30]. At cooling, PCL manages to completely crystallize so that its melting peak shifts towards higher temperatures when reheated. In the case of PLA, the inflection that characterizes the transition to the viscoelastic state is shifted towards lower temperatures. In addition, during further heating, cold crystallization is not explicitly expressed.

### 3.2. Mechanical Properties

The compression test results (Figure 4) show that PCL inclusion in the PLA matrix reduces mechanical characteristics, such as Young’s modulus (*E*) and yield strength, as also shown in Table 1. These values are directly related to the possibility of using this composite material for self-fitting implants, as before installing an implant, for instance, a preliminary load is required to fix the temporary shape. In this case, the stress value should not exceed 80 MPa to avoid irreversible deformations of the material.

### 3.3. Shape Memory Effect

The dynamic mechanical analysis of the samples was performed to determine the temperature phase transitions of a composite blend containing 10 wt.% PCL. Figure 5 shows the graphs of the dynamic elastic modulus, loss modulus, and *tan δ* as a temperature function.

To evaluate SME, the DMA was used to determine the transition temperature that corresponds to the temperature of SME activation. It is characterized by a drop in the elastic modulus at the transition temperature. In this case, there is a peak on the elastic modulus curve, the maximum of which is reached at 45.6 °C. Above this temperature, there is a dramatic fall in the modulus of elasticity. Accordingly, the transition of the PLA/PCL blend to the viscoelasticity starts at a temperature of about 46 °C, which is almost 10 °C lower than for pure PLA.

The peak on the *tan δ* curve indicates an abrupt shift in the strain. In comparison with the pure PLA, this peak is shifted towards lower temperatures. This is probably due to the melting of one of the phases. The PCL melting point varies between 59 and 64 °C, which is shown on the curve.

When studying SME by various methods, such as DSC, where the focus is on thermal transitions in the material, and DMA, where the mechanical characteristics are studied as a function of temperature, there is a discrepancy in temperatures characterizing the same processes in terms of SME. The trigger for starting the SME activation in the PLA/PCL blend is the beginning of the melting process in PCL. At first heating of the DSC, the values in the melting region of PCL and the glass transition of DSC are unclear. This and the apparent peak of 45.6 °C of the storage modulus can be explained by the fact that PCL softens and melts at this set temperature, providing greater mobility of molecular chains. Therefore, the pronounced SME occurs below the glass transition temperature of the PLA matrix.

To study the SME parameters, such as recovery stress and recovery strain, the 3.5 cm long samples were deformed at 100%. Figure 6 shows the recovery strain curve as a function of temperature. For this test, the samples with a fixed temporary shape were placed in grips, one of which was not fixed, and heated from 26 to 70 °C at a rate of 2 °C/min. When the temperature reached 53.3 °C, the samples’ length began to decrease, and the samples returned to the permanent shape.

The same heating parameters were provided for the study of the recovery stress. The samples deformed to the temporary stretched shape were placed in fixed grips. When heated to 40 °C, relaxation occurred within the material. After that, the stress increased up to 1 MPa. When compared with the previous work on SME in PLA composites, it can be seen that recovery stresses increased from 1.5 MPa for neat PLA to 3 MPa in the case of dispersedly filled PLA [19], which was associated with the appearance of an additional rigid “fixed” phase and increased crystallinity. On the contrary, in the current study, the decrease in recovery stresses is associated with the presence of an additional soft phase—PCL.

A significant characteristic of the polymer shape memory effect is the apparent activation energy. It determines the height of the energy barrier that must be overcome to achieve shape recovery. The DMA results were used to determine this characteristic for pure PLA, polymer blend PLA + 10%PCL, and pure PCL. Figure 7 presents the Arrhenius plots of the specific rate constant to determine the activation energy. Table 2 shows the apparent activation energy values. For the glass transition activation of the studied polymer blend PLA + 10%PCL, this value was approximately 186 kJ/mol, 85 kJ/mol less than for pure PLA. As previously assumed, the transition to the viscoelastic state of the polymer blend requires less energy, which means less energy is needed to activate shape memory. The activation energy of the melting processes of pure PCL has a significantly higher energy barrier.

To demonstrate the shape memory effect, a plate was fixed in a temporary shape in the form of a rolled tube (Figure 8A). The plate was heated in a metal container in the air at a temperature of 50 °C. The plate recovered a shape close to the permanent one in 2.5 min, but it did not fully restore its shape due to a lower recovery stress than in the pure PLA. Figure 8B shows the shape memory effect in the PLA/PCL blend.

### 3.4. Biological Compatibility Test

The separately considered polymers of PLA and PCL are biocompatible and are widely used as medical devices, such as fixing rods, plates, pins, screws, and suture materials [15].

To assess the biocompatibility of the polymer composite, the SC1 cell line (7000 cells) was plated in the wells of a 96-cell well. After 48 h of incubation cell cultivation with the samples, a cell viability analysis was performed using spectrophotometry (Figure 9).

Based on the results, it could be determined that the polymer composite material PLA + 10%PCL had no acute toxic effect on the cells. At the same time, there was a slight decrease in the percentage of cell viability compared to the control. This may be due to mechanical damage to the cells when organizing the experiment or to the fact that the cells were deposited in higher numbers at the bottom of the plate than on the surface of the sample.

The modification of the material composition in this study made it possible to adapt the shape memory effect for medical applications, such as self-fitting implants. However, several biological and clinical studies in vitro and in vivo are necessary to fully approve the possibility of using this material. In addition, the reduction of mechanical properties also implies further work with the composition. The incorporation of bioactive ceramic particles into the composition [19] or the creation of block copolymers [36] can positively affect the amount of the rigid phase, improving the mechanical characteristics and the characteristics of the SME. 

## 4. Conclusions

To reduce the SME activation temperature, a polymer blend based on polylactide and polycaprolactone has been suggested. 

The droplet-shaped particles of PCL are distributed in the PLA matrix, which occurs due to their immiscibility. In this case, the inclusions act as a trigger for SME activation due to their melting, making it possible to achieve this temperature. 

The introduction of PCL in the PLA matrix leads to a reduction of mechanical characteristics. Notably, Young’s modulus decreased by almost 200 MPa, which can be related to the efficiency of the SME use.

The transition temperature of 45.6 °C was determined by the DMA method. This temperature corresponds to the temperature of the SME activation. Finally, for the glass transition activation of the investigated polymer blend PLA + 10%PCL, this value was 85 kJ/mol less than for the pure PLA. As previously assumed, the transition to the viscoelastic state of the polymer blend requires less energy, which means less energy is needed to activate shape memory.

The composite material was also examined for biological compatibility. No acute toxic effects on the cells were detected.

By reducing the activation temperature of the shape memory effect to an acceptable use in the body and by reducing the energy barrier of the activation process, the composite material PLA + 10%PCL is a promising material for self-fitting bone implants and other biomedical applications.

## Figures and Tables

**Figure 1 polymers-13-02367-f001:**
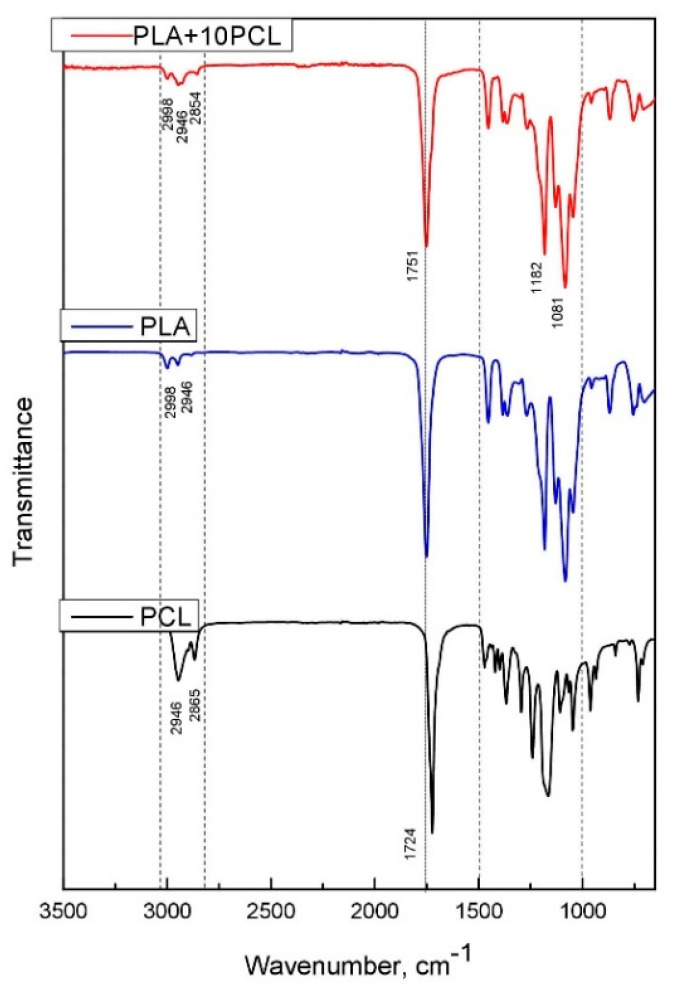
FTIR spectrum of PLA, PCL, and polymer blend PLA+10%PCL.

**Figure 2 polymers-13-02367-f002:**
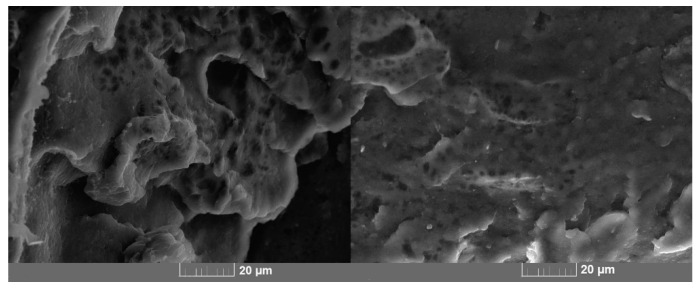
Microphotographs of a quasi-brittle fracture surface of a polymer blend.

**Figure 3 polymers-13-02367-f003:**
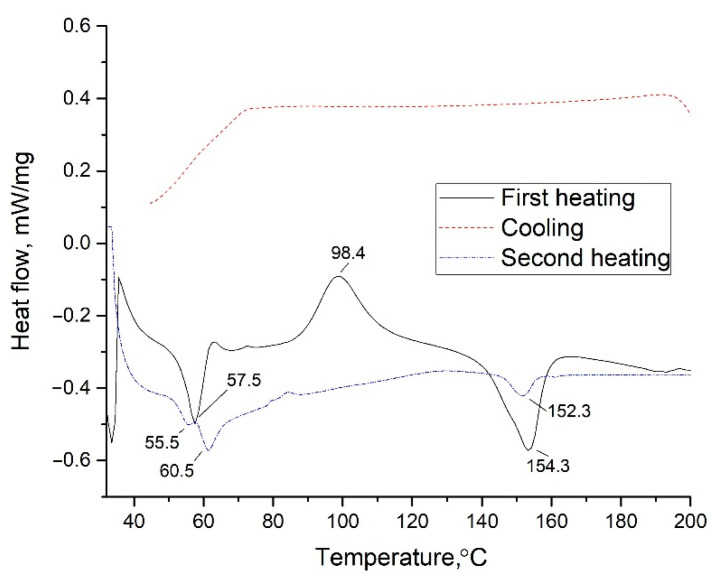
DSC curves for a polymer blend PLA + 10%PCL.

**Figure 4 polymers-13-02367-f004:**
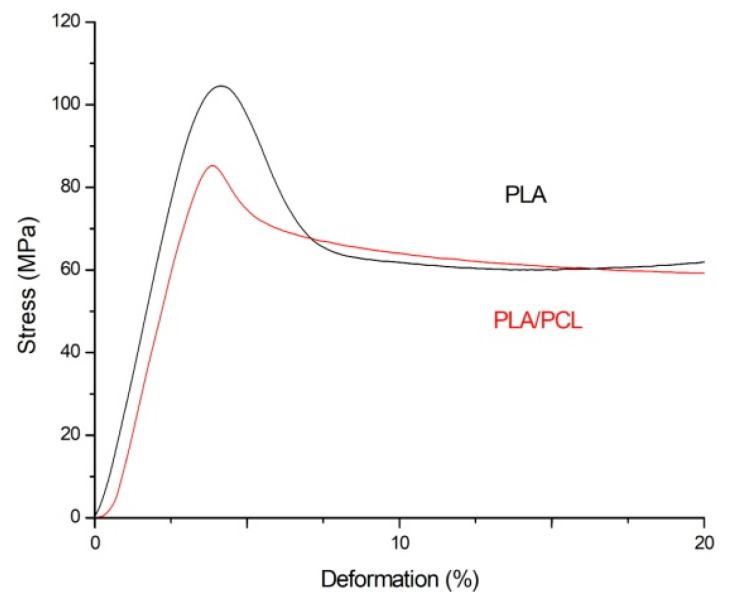
Compression curves for a polymer blend PLA + 10%PCL and PLA.

**Figure 5 polymers-13-02367-f005:**
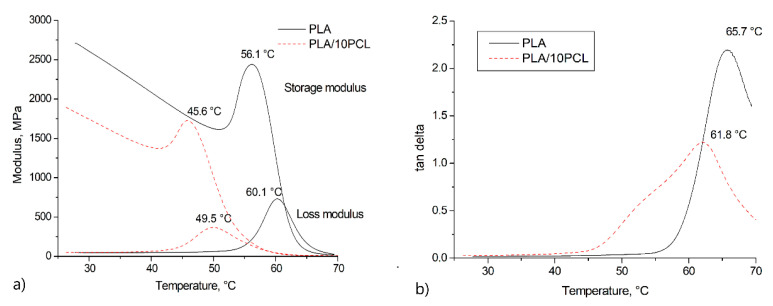
DMA curves of the dynamic elastic modulus, loss modulus (**a**) and tan δ (**b**) for PLA and PLA + 10%PCL.

**Figure 6 polymers-13-02367-f006:**
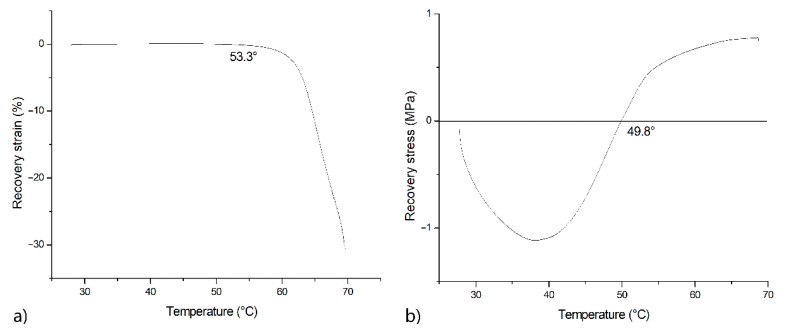
Recovery strain (**a**) and stress (**b**) curves for PLA + 10%PCL.

**Figure 7 polymers-13-02367-f007:**
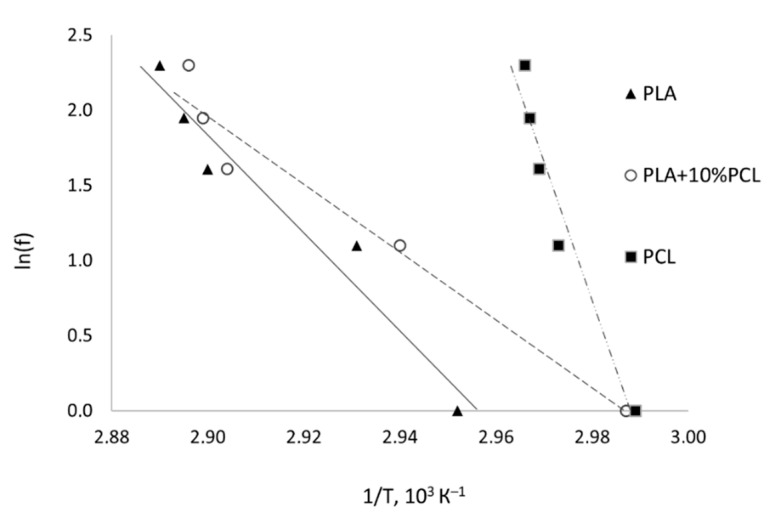
Arrhenius plot of glass transition for the PLA, PLA + 10%PCL, and PCL samples.

**Figure 8 polymers-13-02367-f008:**
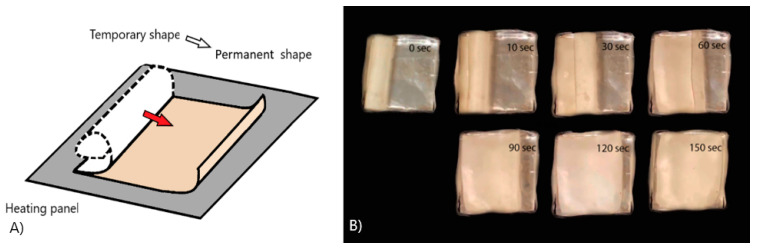
Scheme (**A**) and experiment (**B**) of gradual action of the SME in polymer blend PLA + 10%PCL.

**Figure 9 polymers-13-02367-f009:**
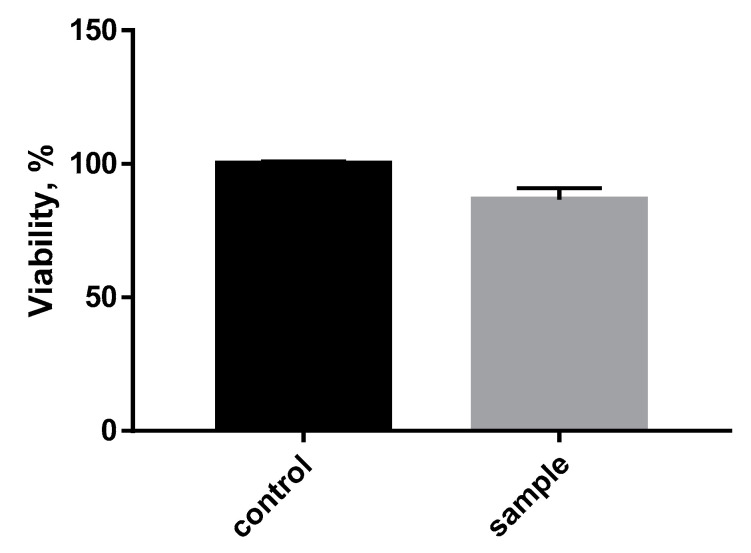
Evaluation of the cytotoxicity of the polymer blend PLA + 10%PCL for a SC1 cell line.

**Table 1 polymers-13-02367-t001:** Results of mechanical compression tests.

	Young’s Modulus, MPa	Yield Strength, MPa
PLA	3295.3 ± 170.0	104.5 ± 2.5
PLA + 10%PCL	3108.4 ± 225.7	85.3 ± 3.9

**Table 2 polymers-13-02367-t002:** Determination of the apparent activation energy (*E_a_*) of glass transition by DMA for the PLA, PLA + 10%PCL, and PCL samples.

Material	*f*, Hz	*T*, °C	*tan δ*	Slope	Correlation Coefficient	Δ*E_a_*, kJ/mol
PLA	1	65.7	2.196	−32.59	0.95	271
	3	68.2	0.637			
	5	71.8	0.705			
	7	72.4	0.701			
	10	73.0	0.692			
PLA + 10%PCL	1	61.8	1.225	−22.50	0.95	186
	3	67.1	0.637			
	5	71.4	0.705			
	7	72.0	0.701			
	10	72.3	0.692			
PCL	1	61.6	0.118	−90.56	0.945	748
	3	63.4	0.159			
	5	63.8	0.140			
	7	64.1	0.126			
	10	64.2	0.098			

## Data Availability

The data presented in this study are available on request from the corresponding author.

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
