# Peer review of "Polymer Composite Materials Based on Polylactide with a Shape Memory Effect for “Self-Fitting” Bone Implants"

_polymers, 2021, doi:10.3390/polym13142367_

Round 1
Reviewer 1 Report
The scientific paper "Polymer composite materials based on polylactide with a shape memory effect for “self-fitting” bone implants" aimed to, in this study, reducing the initiation temperature of the activation and the activation energy of SME was achieved by forming a composite based on PLA containing 10% poly (PCL). It can be considered that:
1) I recommend improving the abstract with more details on methodology and results.
2) Topic 3 should be Results and Discussion.
3) Please correct line 181.
4) Include in vivo study perspectives and limitations in the conclusions.
Author Response
The scientific paper "Polymer composite materials based on polylactide with a shape memory effect for “self-fitting” bone implants" aimed to, in this study, reducing the initiation temperature of the activation and the activation energy of SME was achieved by forming a composite based on PLA containing 10% poly (PCL). It can be considered that:
1) I recommend improving the abstract with more details on methodology and results.
The abstract has been corrected in accordance with the comment, more detailed data has been added.
2) Topic 3 should be Results and Discussion.
There was a mistake in the first version of the manuscript: the Results section was supposed to be combined with the Discussion. The title of the section has been corrected.
3) Please correct line 181.
This section has been corrected in accordance with the comment.
4) Include in vivo study perspectives and limitations in the conclusions.
Following the comment, a cytotoxicity test was included in this work, showing a preliminary assessment of the material's biocompatibility. Also, the perspectives of more substantial biological research in vitro and in vivo were included in the relevant sections.

Reviewer 2 Report
The manuscript titled “Polymer composite materials based on polylactide with a shape memory effect for “self-fitting” bone implants” submitted by Zhukova et al., describes the incorporation of shape memory characteristics in polylactide (PLA) after blending it in different ratio with the plasticizer poly (ε -caprolactone). The topic of work is interesting but addressed extensively by recent literatures such as Naddeo et al., Polymers (Basel). 2021; 13(4): 627. Hence, the present work lacks novelty. The authors are ignorant of relevent literatures like above. In addition, very little attention is paid in the arrangement of references. In text there is Reference 36 while no corresponding reference is present in the reference list.
The authors are too much enthusiastic and claimed that the blended materials as “self-fitting” bone implants without performing any cell culture studies. The preliminary biocompatibility studies must be incorporated for any such claim.
The manuscript lacks the prescribed format (Abstract-Introduction-Materials & Methods- Results-Discussion-Conclusion). There is no Discussion section. The conclusion appears more like discussion than summary of the critical findings.
A good attempt in an interesting topic but could benefit from revisions as described above.
Author Response
The manuscript titled “Polymer composite materials based on polylactide with a shape memory effect for “self-fitting” bone implants” submitted by Zhukova et al., describes the incorporation of shape memory characteristics in polylactide (PLA) after blending it in different ratio with the plasticizer poly (ε -caprolactone). The topic of work is interesting but addressed extensively by recent literatures such as Naddeo et al., Polymers (Basel). 2021; 13(4): 627. Hence, the present work lacks novelty. The authors are ignorant of relevent literatures like above. In addition, very little attention is paid in the arrangement of references. In text there is Reference 36 while no corresponding reference is present in the reference list.
More careful literature review was carried out, adjusted in accordance with the comment. The article indicated turned out to be extremely interesting and close to the subject. It was mentioned in the manuscript and added to the list of references. However, our work is more focused on the study and calculation of the parameters of the shape memory effect.
The authors are too much enthusiastic and claimed that the blended materials as “self-fitting” bone implants without performing any cell culture studies. The preliminary biocompatibility studies must be incorporated for any such claim.
Thank you for the comment. In accordance with the comment, a preliminary study on cytotoxicity was conducted. It confirmed the prospects of the material in the declared area. The text and figures of the manuscript were updated.
The manuscript lacks the prescribed format (Abstract-Introduction-Materials & Methods- Results-Discussion-Conclusion). There is no Discussion section. The conclusion appears more like discussion than summary of the critical findings.
There was a mistake in the first version of the manuscript: the Results section was supposed to be combined with the Discussion. The title of the section has been corrected. In Conclusion, the corrections were also made.

Round 2
Reviewer 2 Report
Thank you, authors, for taking great care of the previous comments. However, the revised manuscript still ignores the major concerns of biocompatibility raised earlier. The SC1 cells used in the revised manuscript are mouse fibroblasts cells. How authors confirm the potentiality of the fabricated material as bone implants by using fibroblasts? Cells act differently with the stiffness of the materials.
It is not clear how the cell seeding was carried out. Please explain whether it is drop-wise seeding or diffusion? What is the volume of the media containing the final number of cells? To confirm the biocompatibility, few weeks cell culture data is required.
How the material was sterilized before seeding?
Please confirm the morphology of the cells after treatment in order to confirm no adverse effect of the material.
Author Response
Thank you, authors, for taking great care of the previous comments. However, the revised manuscript still ignores the major concerns of biocompatibility raised earlier. The SC1 cells used in the revised manuscript are mouse fibroblasts cells. How authors confirm the potentiality of the fabricated material as bone implants by using fibroblasts? Cells act differently with the stiffness of the materials.
The main goal of the work is to develop a polymeric material with shape memory based on polymers that are biocompatible. Using cells, we have shown that the material does not affect the viability of normal cells. Thus, it can be assumed that the use of this material as a bone implant will not have a negative or toxic effect on the organism. This issue will be further studied in the following works.
It is not clear how the cell seeding was carried out. Please explain whether it is drop-wise seeding or diffusion? What is the volume of the media containing the final number of cells? To confirm the biocompatibility, few weeks cell culture data is required.
We thank the Reviewer for this comment. We rewrote a protocol the cell seeding. Answering your questions, the number of cells was 7000 per well, the total volume of growth medium was 200 μl. The sample was placed in the well before the cells were plated. Incubating the sample with cells for a longer time than 2-3 days (for example, a week) while maintaining the normal composition of the growth medium is impossible in this case. In this experiment, we showed the fundamental non-toxicity of the obtained material.
How the material was sterilized before seeding?
We thank the Reviewer for this valuable comment. We added information about sterilization in Materials and Methods section.
Please confirm the morphology of the cells after treatment in order to confirm no adverse effect of the material.
We thank the Reviewer for this comment. The morphology of the cells after treatment you can see in the pictures in the attached file.
